# Milk Consumption Decreases Risk for Breast Cancer in Korean Women under 50 Years of Age: Results from the Health Examinees Study

**DOI:** 10.3390/nu12010032

**Published:** 2019-12-21

**Authors:** Woo-Kyoung Shin, Hwi-Won Lee, Aesun Shin, Jong-koo Lee, Daehee Kang

**Affiliations:** 1Department of Preventive Medicine, Seoul National University College of Medicine, Seoul 03080, Korea; shiningwk@gmail.com (W.-K.S.); hwiwon@snu.ac.kr (H.-W.L.); shinaesun@snu.ac.kr (A.S.); 2Department of Biomedical Sciences, Seoul National University Graduate School, Seoul 03080, Korea; 3Cancer Research Institute, Seoul National University, Seoul 03080, Korea; 4Department of Family Medicine, Seoul National University Hospital, Seoul 03080, Korea; kcdc7000@gmail.com

**Keywords:** milk consumption, breast cancer, cohort study, Korean women, Health Examinees (HEXA) study

## Abstract

Epidemiologic studies regarding breast cancer risk related to milk consumption remain controversial. The aim of this study was to evaluate the association between milk consumption and the risk for breast cancer. A total of 93,306 participants, aged 40–69 years, were included in the prospective cohort study in the Health Examinees-Gem (HEXA-G) study between 2004 and 2013. Dietary intake was assessed using a validated food frequency questionnaire. Information on cancer diagnosis in the eligible cohort was retrieved from the Korea Central Cancer Registry through 31 December 2014. The Cox proportional hazards model was used to estimate multivariate hazard ratios (HRs) and 95% confidence intervals (CIs). A total of 359 breast cancer cases were observed over a median follow-up period of 6.3 years. Milk consumption was not associated with decreased risk for breast cancer in the total population (*p* for trend = 0.0687). In women under 50 years of age, however, milk consumption was inversely associated with breast cancer risk. In the comparison between highest (≥1 serving/day) and lowest (<1 serving/week) intake categories of milk, the multivariate HR (95% CI) was 0.58 (0.35–0.97, *p* for trend = 0.0195)) among women under 50 years of age. In conclusion, our findings show that milk consumption in Korean women aged 50 or younger is associated with a decreased risk for breast cancer, when compared to those who never or rarely consumed milk. Further studies need to be conducted to assess this relationship and confirm these results.

## 1. Introduction 

The breast cancer incidence rate varies greatly worldwide from 25.9 per 100,000 women in South Central Asia to 92.6 per 100,000 women in Western Europe [1]. Although breast cancer incidence rate is lower in Korea than in other countries, it is the most frequently diagnosed type of cancer (19.9%) among women, and the age-standardised incidence of breast cancer has increased in Korea in recent years [2]. However, the five-year relative survival rate of breast cancer was reported to be greater than 90% [3], so understanding the modifiable factors related to breast cancer risk is important.

Reproductive or menstrual factors, a family history of the disease, and the use of hormones are associated with an increased risk for breast cancer [4,5,6,7]. Recently, several studies have shown that breast cancer risk is related to factors that are modifiable by changes in one’s diet and lifestyle. How diet can influence the development of breast cancer has been extensively studied [8,9]. Although an increased risk in women with high alcohol consumption is consistent, epidemiological evidence of the association between specific dietary factors and the risk for breast cancer is limited [10].

Dairy products are an important food group in a diet and have bioactive components that can affect the risk for breast cancer. The consumption of milk and/or dairy products, due to the high saturated fat content and insulin-like growth factor 1 (IGF-1), has been hypothesized to influence the risk for breast cancer [11,12]. In contrast, the anti-carcinogenic properties of some dairy-related nutrients such as calcium, vitamin D, and conjugated linoleic acids, have been proposed as potential mechanisms conferring a protective effect of milk and dairy products resulting in a reduction in breast cancer risk [13,14,15,16]. 

Several epidemiologic studies have examined the association between consumption of milk and dairy products and the risk for breast cancer [17,18,19]. However, the association between different types of milk and dairy products and risk for breast cancer yield inconsistent results, as only a few studies have shown reduced risks for breast cancer [20,21,22]. Moreover, the findings may vary by the types of dairy products.

Therefore, we examined the association between milk and dairy consumption and the risk for breast cancer among Korean women in the Health Examinees-Gem (HEXA-G) study.

## 2. Method

### 2.1. Study Population

The Health Examinees (HEXA) study recruited participants aged 40–69 years between 2004 and 2013 from 38 general hospitals and health examination centers in eight regions around Korea in accordance with standardized study protocol. Well-trained interviewers used a structured questionnaire to collect information on socio-demographic and lifestyle factors, disease history, and dietary habits. Information on reproductive history was also collected from the women, and trained medical staff performed physical examinations on all participants. Details about the HEXA study are described elsewhere [23,24]. The HEXA study is within the Korean Genome and Epidemiology Study (KoGES). 

Updated from the previously published HEXA studies [23,24], the current study used the HEXA-G participant sample which includes additional eligibility criteria on the participating sites. Exclusion criteria for the recruiting center are described in a previous study [25]. In the HEXA-G study, 93,306 women were included at baseline. 

Among the HEXA-G cohort, we excluded participants who had no information on the date and cause of cancer from the Korea Central Registry (*n* = 8531) and those who did not provide information on food frequency questionnaire (FFQ) (*n* = 1165). Missing data on FFQ was treated according to the criteria provided by KoGES, which excluded individuals who: (1) did not answer any of the questions in the FFQ; (2) left more than 12 frequency questions blank; (3) did not answer any questions on rice consumption; or (4) had extremely low (<100 kcal/day) or high (≥10,000 kcal/day) energy consumption [26]. Participants were also excluded if they did not have plausible values for total energy consumption (*n* = 1069; <500 or >3500 kcal/day).

We excluded women who were diagnosed with cancer before enrolment (*n* = 4000) and women who were diagnosed with breast cancer during the first two years of the follow-up period to minimize the impact of reverse causation (*n* = 221). As a result, our study included a total of 78,320 participants. 

The study protocol was approved by the Ethics Committee of the KoGES of the Korean National Institute of Health. All procedures for this study were approved by the Institutional Review Board (IRB) of the Seoul National University Hospital in Seoul, Korea (IRB number 0608-018-179) and the Korea National Institute of Health (IRB number 2014-08-02-3C-A). Written informed consent was obtained from all participants.

### 2.2. Dietary Assessment

The usual diet from the previous year was assessed using an interviewer-administered semi-quantitative FFQ with 106 food items [27,28]. The frequency of servings was classified into nine categories: almost never, once a month, two or three times a month, once or twice a week, three or four times a week, five or six times a week, once a day, twice a day, or three times or more a day. The portion size was categorized as small, medium, or large. 

Dairy products included in the FFQ items included milk (1 cup, 200 mL), yogurt (1 serving, 120 mL), and cheese (1 slice, 20 g) [26]. The amount of dairy product consumption was converted to daily frequencies and then multiplied by the reported portion sizes for each food item. 

Total energy and nutrient intakes were calculated by multiplying the reported frequency of each food item by the nutrient content in a serving of that food. The nutrient values were based on the food composition table in the Recommended Dietary Allowances for Koreans [29]. 

The reliability and validity of the FFQ is described in a previous study [28]. The FFQ was validated by comparing nutrient intake and foods obtained from the second FFQ with those derived from the 12-day dietary records. A de-attenuated correlation coefficient with adjustments for age, sex, and energy intake ranged between 0.23 (vitamin A) and 0.64 (carbohydrate). The reliability of the FFQ was assessed by comparing the intake of nutrients obtained from the two FFQs that were administered one-year apart. The median correlations between the two FFQs were 0.45 for all nutrient intake values and 0.39 for all nutrient densities (nutrient/energy). The correlation coefficients for nutrient intake varied from 0.24 (carbohydrate) to 0.58 (cholesterol). 

Consumptions of carbohydrates (g/day), protein (g/day), fat (g/day), fruits (g/day), and vegetables (g/day) were total energy intake adjusted using the residual method [30].

### 2.3. Identification of Breast Cancer

Information on cancer diagnoses in the eligible cohort was obtained by linkage to the Korea Central Cancer Registry, using resident registration number, a unique number assigned to every citizen and foreign resident in Korea. Cancer incidence until 31 December 2014 was provided by the National Cancer Center Korea. Breast cancer was defined based on the 10th revision of the International Classification of Disease (ICD-10) code C50. The primary outcome was breast cancer incidence. The first cancer diagnosis was considered, resulting in 359 breast cancer cases identified and a total of 78,320 participants remained in the final analysis (Figure 1).

### 2.4. Demographic and Anthropometric Measures

Each participant’s age, educational level, age at first birth, number of live births, age at menarche, oral contraceptive use, exercise stratus, alcohol consumption, smoking status, marital status, and menopausal status were obtained via structured questionnaire.

We obtained information on the participants’ physical examinations including their height (cm), weight (kg), and waist circumference (cm) directly measured by the medical staff using the standardized KoGES protocol of Korea Centers for Disease Control and Prevention. Body mass index (BMI) (kg/m^2^) was calculated by dividing the body weight in kilograms by the height in meters squared.

### 2.5. Statistical Analysis

Participants were classified into three or four categories based on the distribution of their reported daily intake of each targeted milk and dairy food item. As the number of participants who never or rarely consumed (<1 serving/week) milk (*n* = 21,242, 27.12%) or yogurt (*n* = 24,380, 31.13%) was less than those who never or rarely consumed cheese (*n* = 57,074, 72.87%), we categorized these women into three (<1 serving/week, 1 serving/week, or ≥2 servings/week, for cheese consumption) or four (<1 serving/week, 1 serving/week, 2–6 servings/week, or ≥1 serving/day, for milk and yogurt consumption) groups.

Descriptive analyses were performed using the Chi-squared test and analysis of variance (ANOVA) and the means and standard deviations for continuous variables and the number and percentages for categorical variables were provided.

Hazard ratios (HRs) and 95% confidence intervals (CIs) were calculated by Cox proportional hazards models. Since age is one of the most important determinant for cancer outcome, we chose age as the time scale in Cox proportional hazard models. Age at baseline was used as the entry time to reflect the left-truncation time of the data [31]. Age at cancer diagnosis, time of censoring or last follow-up was the exit time. 

Person-years of follow-up were calculated from the date the baseline questionnaire was returned to the date of breast cancer diagnosis, date of cancer diagnosis, or end of follow-up (31 December 2014), whichever came first.

Multivariate HRs were adjusted for BMI (kg/m^2^, continuous), and energy intake (kcal/day, continuous), educational level (middle school or less, high school or college, undergraduate or more), parity (nulliparous, 1, 2, ≥3), age at first birth (nulliparous, aged < 25 years at first birth, aged ≥25 at first birth), age at menarche (< 15, 15, ≥16 years), oral contraceptive use (never, ever), regular exercise (no, yes), alcohol consumption (never, ever), and family history of breast cancer (no, yes). To test for linear trends across categories of milk and dairy product intake, participants were assigned the median value of each category, and this variable was entered as a continuous term in the model, the coefficient for which was evaluated by the Wald test. 

We evaluated whether the association varied by age (ages 40–49 or ≥50 years). The likelihood ratio test was used to test for significance in interaction models by comparing models with and without cross-product terms.

The proportion of participants with missing data for the covariates was generally low (<3%); we thus assigned median values for continuous variables and the most common category for categorical variables to the missing variables in the participant data.

To examine the possible presence of a time lag effect, we excluded the first two years of the follow-up period from the analysis.

The statistical software package SAS for Windows version 9.4 (SAS Institute, Cary, NC, USA) was used for all statistical analyses and two-side tests were statistically significant at *p* ≤ 0.05.

## 3. Results

A total of 78,320 participants, aged 40 to 69 years, were included in the analysis. During an average follow-up period of 6.3 years, 359 women were diagnosed with breast cancer through 31 December 2014. The mean age of the participants was 52.33 years.

Table 1 presents the demographic characteristics and dietary intake according to amount of milk consumption. Women who consumed more than a single serving of milk daily were younger, less obese, married or were cohabitants, highly educated, physically inactive, non-smokers, and ever drinkers compared to women who consumed less than one serving of milk per week. Individuals with the highest consumption of milk were more likely to have higher total energy intake, higher protein intake, and lower fat and carbohydrate intake than those who consumed less than one serving per week.

The multivariate HRs (95% CIs) of breast cancer according to milk consumption are shown in Table 2. Milk consumption was not associated with a decreased risk for breast cancer in the total population (*p* for trend = 0.0687). In women under 50 years of age, however, milk consumption was inversely associated with breast cancer risk. In the comparison between the highest (≥1 serving/day) and lowest (<1 serving/week) intake categories for milk, the multivariate HR (95% CI) was 0.58 (0.35–0.97), and increased consumption of milk was associated with a lower risk for breast cancer (*p* for trend = 0.0195) among women under 50 years of age. However, this interaction was not statistically significant.

We observed no association between breast cancer risk and yogurt or cheese consumption in either age groups (Appendix A).

## 4. Discussion

In this prospective cohort of Koreans, we found that milk consumption in women aged 50 or younger was associated with a decreased risk for breast cancer, when compared to those who never or rarely consumed milk (<1 serving/week). These results are in line with findings from previous studies, which also found association between decreased risk for breast cancer and high consumption of milk [18,32]. A prospective cohort of 88,691 women in the Nurses’ Health Study reported that a high intake of skim/low-fat milk was associated with a reduced risk for breast cancer among premenopausal women [18]. According to the French SUpplémentation en VItamines et Minéraux AntioXydants (SU.VI.MAX) prospective study, dairy products, due to calcium content or a correlated component, might have a negative association with the risk for breast cancer, particularly among premenopausal women [22]. A case-control study indicated that a diet characterized by a high consumption of milk is associated with a lower risk for breast cancer among both pre- and postmenopausal women in China [33]. Meta-analyses reported that dairy product consumption was inversely associated with the risk for breast cancer, and this effect varied across types of dairy products [20,34,35]. 

Previous studies reported that dairy products have anti-carcinogenic effects, due to large amounts of calcium, vitamin D, and some bioactive components including conjugated linoleic acid (CLA) [17,18,22]. It has been suggested that calcium protects against breast carcinogenesis because of its anti-proliferative properties, which promote processes such as the regulation of cell proliferation and differentiation [13]. Lactose can aid the absorption of dietary calcium as well as promote the growth of lactic acid, leading to the production of bacteria in the large intestine. Lactose has been hypothesized to increase ovarian cancer risk by direct toxicity to oocytes and by inducing premature ovarian failure [36]. This effect can reduce the exposure of breast tissue to estrogen. Vitamin D has also been shown to interrupt insulin and IGF-1, which may lower carcinogenic risk as insulin stimulates a rise in free IGF-1, which may promote cell cycle progression and angiogenesis and is anti-apoptotic [37,38]. CLA, a mixture of positional and geometric isomers of linoleic acid, comes from dairy products [39] and is a potent anticarcinogen [40]. In this study, we did not find any statistically significant associations between yogurt or cheese consumption and the risk for breast cancer in either age groups. Consistent with our research, some prospective cohort studies showed no association between yogurt or cheese consumption and breast cancer risk. In the Black Women’s Health Study, there were no associations between the intake of specific types of dairy products and breast cancer risk [41]. In a pooled analysis of eight prospective studies, null associations were observed between yogurt products, cottage cheese, and cheese products and the risk for breast cancer [42]. 

Several explanations can be postulated that would explain our inconsistent findings on milk and other dairy product consumption. Dairy products have both pro- and anticarcinogenic effects. Dairy products, which are relatively high in calcium, vitamin D, and CLA, have effects on cell proliferation and differentiation and/or inhibit tumor development [13,14,15,16]. On the contrary, high amounts of fat, saturated fat and potentially carcinogenic contaminants (pesticides, oestrogen metabolites, and growth factors including IGF-1) in dairy products can increase the risk for breast cancer [14,43,44]. 

Our findings showed that milk consumption was not associated with a decreased risk in the study population as a whole. When stratified by age group, however, an inverse association, was observed in women less than 50 years of age, whereas among older women the association was null. Since some risk factors for breast cancer vary according to age or menopausal status and the etiologies of pre or postmenopausal breast cancer are different in many aspects, the association between diet and breast cancer risk may be different between younger vs. older women and pre vs. postmenopausal women [9]. Consistent with our research, several previous studies have reported that the association between dairy product consumption and the risk of breast cancer was stronger in premenopausal women than in postmenopausal women [18,22,45]. A potential explanation was a dilution of association between dairy product intake and the risk for postmenopausal breast cancer by use of hormone replacement treatment [35]. In addition, the interaction among calcium, vitamin D, and IGF-1 may be stronger for premenopausal women than for postmenopausal women [45]. Previous studies have reported that the association of milk intake and related nutrients with a reduced risk of breast cancer was null after menopause [18] or with increasing age [46].

Potential limitations need to be considered in our study. First, measurement errors in dietary assessments based on self-reported FFQs are unavoidable. However, intake assessments of nutrients based on our FFQs have reasonable validity and reproducibility [28]. FFQs are the most frequently used assessment tools in large-scale epidemiological studies due to their cost effectiveness and the short amount of time required to administer. Also, information bias by differential recall was minimized because dietary assessment was performed at the baseline in this study when participants were free of cancer and unaware of the later research hypothesis. Second, we could not obtain information on the types of milk (whole, low-fat, and skim) from our FFQs because few adults consume low-fat or skim milk in Korea. Third, we could not obtain information on clinical data, including estrogen and progesterone receptor (ER/PR) status for breast cancer patients. Further prospective studies that control for clinical factors need to be conducted to evaluate the association between dietary factors and breast cancer risk. Finally, the duration of the follow-up period and number of cases were limited, especially in the analyses stratified by breast cancer risk factors.

Despite these limitations, the strengths of our study include prospective design, large population, and high accuracy of breast cancer diagnoses, as the data on cancer diagnoses in this cohort were retrieved from the Korea Central Registry. Moreover, information bias by differential recall was minimized because the dietary assessments were performed at baseline survey when the participants were free of cancer and unaware of the research hypothesis. Additionally, we were able to adjust for reproductive or menstrual factors, presence of breast cancer family history, and hormone use associated with breast cancer risk.

In conclusion, our results provide evidence that milk consumption in Korean women less than 50 years of age is associated with a decreased risk of breast cancer compared to those who never or rarely consumed milk. To confirm any potential effect of specific dairy products, more prospective studies need to be conducted. 

## Figures and Tables

**Figure 1 nutrients-12-00032-f001:**
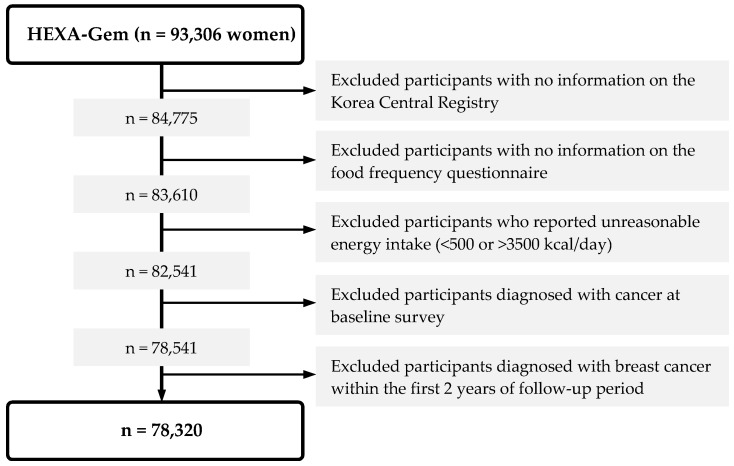
Study population. HEXA, Health Examinees.

**Table 1 nutrients-12-00032-t001:** Characteristics of participants according to milk consumption.

		Milk Consumption (Serving)				
Variables ^a^	Total	<1/week	1/week	2–6/week	≥1/day	*p*-Value ^b^
Number of Participants	78,320	33,787	12,368	13,889	18,276	
Age (years)	52.3 (7.8)	52.7 (7.9)	51.4 (7.7)	51.5 (7.6)	52.9 (7.7)	<0.0001
Body Mass Index (BMI) (kg/m^2^)	23.6 (2.9)	23.7 (3.0)	23.6 (3.0)	23.5 (2.8)	23.6 (2.9)	<0.0001
Educational Level						<0.0001
Middle School or Less	28,465 (36.3)	13,704 (40.6)	4089 (33.1)	4314 (31.1)	6358 (34.8)	
High School or College	33,585 (42.9)	13,879 (41.1)	5596 (45.3)	6228 (44.8)	7882 (43.1)	
Undergraduate School or More	15,405 (19.7)	5,814 (17.2)	2568 (20.8)	3183 (22.9)	3840 (21.0)	
Marital Status						<0.0001
Married or Cohabiting	67,753 (86.5)	29,240 (86.5)	10,828 (87.6)	12,050 (86.8)	15,635 (85.6)	
Single or Divorced or Widowed	10,283 (13.1)	4413 (13.1)	1499 (12.1)	1784 (12.8)	2587 (14.2)	
Regular Exercise						<0.0001
No	38,421 (49.1)	18,014 (53.3)	6176 (50.0)	6301 (45.4)	7930 (43.4)	
Yes	39,717 (50.7)	15,686 (46.4)	6176 (50.0)	7548 (54.4)	10,307 (56.4)	
Alcohol						<0.0001
Never	52,410 (66.9)	23,177 (68.6)	8038 (65.0)	8908 (64.1)	12,287 (67.2)	
Ever	25,619 (32.7)	10,482 (31.0)	4298 (34.8)	4920 (35.4)	5919 (32.4)	
Smoking						0.0184
Never	75,217 (96.0)	32,385 (95.9)	11,925 (96.4)	13,363 (96.2)	17,544 (96.0)	
Ever	2809 (3.6)	1267 (3.8)	412 (3.3)	464 (3.3)	666 (3.6)	
Total Energy Intake (kcal/day)	1981.0 (487.6)	1557.3 (453.9)	1662.3 (458.6)	1752.9 (482.1)	1867.9 (501.6)	<0.0001
Protein (g/day) ^c^	56.8 (10.9)	54.6 (11.0)	56.2 (10.2)	58.1 (10.4)	60.2 (10.4)	<0.0001
Fat (g/day) ^c^	25.8 (10.1)	22.8 (9.9)	25.7 (9.8)	27.8 (9.6)	29.9 (9.3)	<0.0001
Carbohydrate (g/day) ^c^	301.6 (28.4)	309.5 (27.7)	302.6 (27.1)	296.6 (27.3)	290.3 (26.9)	<0.0001
Fruit (g/day) ^c^	236.3 (207.5)	229.4 (217.8)	236.7 (204.1)	239.2 (199.6)	246.7 (195.7)	<0.0001
Vegetable (g/day) ^c^	120.9 (93.4)	118.8 (97.3)	119.0 (88.5)	123.5 (89.4)	124.2 (91.7)	<0.0001

^a^ Continuous variables are reported as mean (standard deviation) values and categorical variables are reported as number (percentage, %). ^b^ Analysis of variance was used for continuous variables and Chi-squared test was used for categorical variables. ^c^ Total energy intake was adjusted using the residual method.

**Table 2 nutrients-12-00032-t002:** Hazard ratios (HRs) and 95% confidence intervals (CIs) of breast cancer according to milk consumption.

	Milk Consumption (Serving)				
	<1/week	1/week	2–6/week	≥1/day	*p* for Trend ^b^
**Total**					
Person-Years	213,016	77,694	88,995	114,786	
Number	33,787	12,368	13,889	18,276	
Breast Cancer Cases	159	61	65	74	
HR (95% CI) ^a^	1.00 (ref)	1.01 (0.75, 1.36)	0.90 (0.67, 1.21)	0.78 (0.59, 1.04)	0.0687
**Age <50 years**					
Person-Years	80,703	33,944	38,095	40,295	
Number	12,464	5286	5792	6261	
Breast Cancer Cases	66	34	29	21	
HR (95% CI) ^a^	1.00 (ref)	1.19 (0.78, 1.80)	0.87 (0.56, 1.36)	0.58 (0.35, 0.97)	0.0195
**Age ≥50 years**					
Person-Years	132,314	43,749	50,900	74,491	
Number	21,323	7082	8097	12,015	
Breast Cancer Cases	93	27	36	53	
HR (95% CI) ^a^	1.00 (ref)	0.84 (0.55, 1.30)	0.93 (0.63, 1.37)	0.90 (0.64, 1.28)	0.6587

^a^ HR (95% CI): adjusted for BMI (kg/m^2^, continuous), total energy intake (kcal/day, continuous), educational level (middle school or less, high school or college, undergraduate or more), parity (nulliparous, 1, 2, ≥3), age at first birth (nulliparous, aged <25 at first birth, aged ≥25 at first birth), age at menarche (<15, 15, ≥16 years), oral contraceptive use (never, ever), regular exercise (no, yes), alcohol consumption (never, ever), and the presence of a family history of breast cancer (no, yes); ^b^
*p* for trend was calculated using the median value of each category as a continuous variable.

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
