# Peer review of "Milk Consumption Decreases Risk for Breast Cancer in Korean Women under 50 Years of Age: Results from the Health Examinees Study"

_nutrients, 2019, doi:10.3390/nu12010032_

Round 1

Reviewer 1 Report

The manuscript entitled “Milk consumption decreases risk for breast cancer in women under 50 years of age: Results from the Health Examinees study” submitted for revision in Nutrients had been positively reviewed with some minor modifications.

This is an interesting manuscript concerning the evaluate the association between milk consumption and the risk for breast cancer.

The review is positive.

Title – I suggest correct and add “Korean women”: Milk consumption decreases risk for breast cancer in Korean women under 50 years of age: Results from the Health Examinees study Keywords should not be the same as in the title of the manuscript - other words should be added The statement: ”Milk consumption was associated with a marginally significantly decreased risk for breast cancer (p for trend = 0.07)” is incorrect because p=0.07 means that there are no significant differences.

In the abstract (Line 19-20) and Results (Line 178 – 179) I suggest change: Milk consumption was not associated with decreased risk for breast cancer (p for trend = 0.0687) in the total population.

Author Response

1. Title – I suggest correct and add “Korean women”: Milk consumption decreases risk for breast cancer in Korean women under 50 years of age: Results from the Health Examinees study

We have revised title as follows;

Milk consumption decreases risk for breast cancer in Korean women under 50 years of age: Results from the Health Examinees study

2. Keywords should not be the same as in the title of the manuscript - other words should be added:

We have revised key words as follows;

Keywords: Milk consumption; Breast cancer; Cohort study; Korean women; the Health Examinees (HEXA) study

3. The statement: ”Milk consumption was associated with a marginally significantly dereased risk for breast cancer (p for trend = 0.07)” is incorrect because p=0.07 means that there are no significant differences. In the abstract (Line 19-20) and Results (Line 178 – 179) I suggest change: Milk consumption was not associated with decreased risk for breast cancer (p for trend = 0.0687) in the total population.

We have revised the manuscript.

Page 1;

Abstract: Milk consumption was not associated with decreased risk for breast cancer in the total population (p for trend = 0.06877).

Page 5, Line 182;

Result: Milk consumption was not associated with decreased risk for breast cancer in the total population (p for trend = 0.0687).

Page 6, Line 239;

Discussion: Our findings showed that milk consumption was not associated with decreased risk, but this association was stronger in young adults less than 50 years of age.

Reviewer 2 Report

Dear Authors

The paper presented is interesting and could represent an important line of research as a means to promote dairy consumption among the young female population. I encourage you to follow this line

Author Response

Thank you for your great evaluation!

Reviewer 3 Report

The title reflects the subject of the study. This manuscript presents a clear and clinically useful message. It is well written in terms of clarity, style, and use of English language. Materials and methods are sufficiently detailed. The discussion section explains adequately the purpose of this study in the context of published information. The conclusions accurately and clearly explain the main results. The length of the manuscript is ideal. All tables are of good quality and relevant to the subject. All references are appropriate and current.

Author Response

Thank you for your great evaluation!

Reviewer 4 Report

This is an epidemiology study on dairy product and the risk of breast cancer in a Korean prospective study of more than >78320 women (359 breast cancer). They found that higher milk consumption was associated with reduced risk of breast cancer in women < 50 years old. The inverse association in those older than 50 years old was not significant. This is an ordinary study without significant novelty. The study seems well-conducted and limitations are discussed. 

Comments below

When making the conclusion, please specify what is the reference group and what is the comparison group Of milk intake. 

Please rephrase the sentence in lines 49 to 50. 

The criterion of extreme energy seems too loose for women. Should it be <500 Kcal and > 3000 kcal for women? Please double check. 

Lines 56, Why didn't answer the question on rice consumption is an exclusion criterion? 

Line 102, Please gives a reference. 

In the abstract, the breast cancer cases is 359. In line 117, the number is 329. Figure 1 seems not referred in the context. 

What is the test for P Trend?

All alcohol should be categorized as in never, former, light drinker, and heavy drinker. The current categorization may have residual compounding concern from light alcohol drink.

In table 1 what is I/day less and equal. P value doesn't provide much information in this study due to the large sample size. 

In table 2. Please provide a follow up person year, rather than the number.

This study is not a particularly novel because other studies have shown the inverse association between milk consumption and a breast cancer risk.

The data for cheese and yogurt can be included as a supplemental table. 

Can the author comment on why the inverse association between milk intake and breast cancer risk was not significant in older woman. 

What is the point the paragraph 4 in discussion the authors like to convey?

Author Response

1. When making the conclusion, please specify what is the reference group and what is the comparison group of milk intake.

We have revised in the manuscript (Abstract and Discussion)

Page 1, Line 24;

Abstract: In conclusion, we found that in women under 50 years of age milk consumption was associated with decreased risk compared to the women who never or rarely consumed milk (<1 serving/week) for breast cancer.

Page 6, Line 201;

Discussion: In this prospective cohort, we found that in women under 50 years of age milk consumption was associated with decreased risk compared to the women who never or rarely consumed milk (<1 serving/week) for breast cancer.

2. Please rephrase the sentence in lines 49 to 50.

We have revised the sentence as follows;

Page 1, Line 41;

Although an increased risk in women with high alcohol consumption was consistent, epidemiological evidence of the association between specific dietary factors and the risk for breast cancer is limited [10].

3. The criterion of extreme energy seems too loose for women. Should it be <500 Kcal and > 3000 kcal for women? Please double check.

The FFQ guideline for Korean Genome and Epidemiology Study (KoGES) of Korea CDC suggested the criterion of extreme energy range to be <500 and >3500 kcal/day. According to this guideline, we thought that the proportion of excluded subjects (n=1,069 (about 1% of the total population), <500 and >3500kcal/day) was reasonable, and thus applied the criterion (<500 and >3500 kcal/day).

Reference : KCDC Center for Genome Sciences. FFQ guideline for Korean Genome and Epidemiology Study (KoGES). 2014. Available online: (https://www.cdc.go.kr/board.es?mid=a20602010000&bid=0034&act=view&list_no=63735 (accessed on 13 December 2019).

4. Lines 56, Why didn't answer the question on rice consumption is an exclusion criterion?

The FFQ guideline for Korean Genome and Epidemiology Study (KoGES) of Korea CDC provided exclusion criteria on various food items including rice consumption.

Because Koreans eat rice as a staple, so most participants are supposed to answer any of the questionnaires related to rice items

Reference : KCDC Center for Genome Sciences. FFQ guideline for Korean Genome and Epidemiology Study (KoGES). 2014. Available online: (https://www.cdc.go.kr/board.es?mid=a20602010000&bid=0034&act=view&list_no=63735 (accessed on 13 December 2019).

5. Line 102, Please gives a reference.

We have added references as follws;

Page 3, Line 91;

The usual diet from the previous 1 year was assessed using an interviewer-administered semi-quantitative FFQ with 106 food items [27,28]. Dairy products included in the FFQ items included milk (1 cup, 200 mL), yogurt (1 serving, 120 mL), and cheese (1 slice, 20 g) [26].

6. In the abstract, the breast cancer cases is 359. In line 117, the number is 329. Figure 1 seems not referred in the context.

359 is right. 329 is typo.

7. What is the test for P Trend?

We have revised the manuscript as follows

Page 4, Line 157;

To test for trend, participants were assigned the median value of each category, and this variable was entered as a continuous term in the model, the coefficient for which was evaluated by the Wald test.

8. All alcohol should be categorized as in never, former, light drinker, and heavy drinker. The current categorization may have residual compounding concern from light alcohol drink.

The current categorization was dichotomous (never / ever). Alcohol drinking frequency and amount in our study subjects was very low. Thus, we had to made only two groups.

9. In table 2. Please provide a follow up person year, rather than the number.

We have added the follow up person-year according to each category in Table 2.

10. This study is not a particularly novel because other studies have shown the inverse association between milk consumption and a breast cancer risk.

Although the few study suggested the inverse association between milk and breast cancer risk (Shin et al., 2002; Kesse-Guyot et al., 2007; Zhang et al., 2011), the strengths of this study include its prospective study design, the large population and high accuracy of the breast cancer diagnoses, as the data on cancer diagnoses in this cohort were retrieved from the Korea Central Registry.

Shin, M.-H.; Holmes, M.D.; Hankinson, S.E.; Wu, K.; Colditz, G.A.; Willett, W.C. Intake of dairy products, calcium, and vitamin D and risk of breast cancer. J. Natl. Cancer Inst. 2002, 94, 1301-1310.

Kesse-Guyot, E.; Bertrais, S.; Duperray, B.; Arnault, N.; Bar-Hen, A.; Galan, P.; Hercberg, S. Dairy products, calcium and the risk of breast cancer: results of the French SU. VI. MAX prospective study. Ann. Nutr. Metab. 2007, 51, 139-145.

Zhang, C.-X.; Ho, S.C.; Fu, J.-H.; Cheng, S.-Z.; Chen, Y.-M.; Lin, F.-Y. Dietary patterns and breast cancer risk among Chinese women. Cancer Causes Control 2011, 22, 115-124.

11. The data for cheese and yogurt can be included as a supplemental table.

We have added supplemental tables.

Supplemental table 1. Hazard Ratios (HR)and 95% Confidence Intervals (CI) of Breast Cancer according to Yogurt Consumption

Supplemental table 2. Hazard Ratios (HR)and 95% Confidence Intervals (CI) of Breast Cancer according to Cheese Consumption

12. Can the author comment on why the inverse association between milk intake and breast cancer risk was not significant in older woman?

We have added why the inverse association between milk intake and the risk for breast cancer was not significant in older women in the manuscript as follows;

Page 8, Line 244;

In consistent with our research, several previous studies have reported that the association between dairy product consumption and risk of breast cancer was stronger in premenopausal women than that in postmenopausal women [18,22,43]. A potential explanation was a dilution of association between dairy product intake and the risk for postmenopausal breast cancer by use of hormone replacement treatment [33]. In addition, the interaction among calcium, vitamin D, and insulin-like growth factors may be stronger for premenopausal women than for postmenopausal women [41]. 

13. What is the point the paragraph 4 in discussion the authors like to convey?

We have revised the manuscript. We have added as follows;

Page 8, Line 232;

Several explanations can be postulated that would explain our inconsistent findings of milk and other dairy product consumption. Dairy products have been hypothesized to have both the pro- and anticarcinogenic effects. Dairy products, which are relatively high in calcium, vitamin D, and CLA, have effects on cell proliferation and differentiation and/or inhibit tumour development [13-16]; on the contrary, dairy products can increase the risk for breast cancer due to the high amounts of fat and saturated fat and potentially carcinogenic contaminants, such as pesticides, oestrogen metabolites, and growth factors such as insulin like growth factor 1 [14,42,43]

Reviewer 5 Report

To,

Editor

Nutrients

Woo-Kyoung Shin and group have shown in this manuscript that intake of milk decreases risk of Breast cancer in middle aged women. Study was performed on 93306 patients by collecting data on various parameters which were used to deduce conclusions. It is an extensive approach undertaken by authors. Authors use standard set of questionnaires to get information about dietary uptakes, health history etc. Data provided by authors support their claim however some of the data is not statistically significant. However, considering large set of patients and various aspects of study to be analyzed, such significance can be understood. The results presented in this manuscript are interesting and will expand our knowledge about breast cancer risks. Despite these qualities of the manuscript, I just have a few points that I request authors to address or discuss it in manuscript.

How frequently data were collected from individual patients? Why only patients excluded due previous diagnosis of breast cancer? Were there any other patients with history of other types cancers? If yes were, they taken for study and why? Was data collection from patients continued till they were diagnosed with breast cancer? Were there any other types of cancer reported in these patients? Since Milk and its derivatives/ingredients have anti-cancer properties, was there any impact on other cancers if data was collected for same?

Author Response

1. How frequently data were collected from individual patients?

Baseline survey for all cohort members was conducted between 2004 and 2013. Information on cancer diagnoses of cohort member was obtained by the Korea Central Cancer Registry in December 2014.

2. Why only patients excluded due previous diagnosis of breast cancer? Were there any other patients with history of other types cancers? If yes were, they taken for study and why?

We have excluded women who were diagnosed with breast cancer during the first 2 years of the follow-up period to minimize the impact of reverse causation (n=221).

We have excluded participants who had a diagnosis of breast cancer and other type of cancer before enrolment at the baseline (n=4000).

3. Was data collection from patients continued till they were diagnosed with breast cancer? Were there any other types of cancer reported in these patients?

We collect from individual patients between 2004 and 2013 at baseline and excluded participants who had a diagnosis of other type of cancer before enrolment at the baseline.